# Direct laser-written optomechanical membranes in fiber Fabry-Perot cavities

Lukas Tenbrake [1,4], Alexander Faßbender[2,4], Sebastian Hofferberth [1], Stefan Linden [2] & Hannes Pfeifer [1,3] ✉

Integrated micro- and nanophotonic optomechanical experiments enable the manipulation of mechanical resonators on the single phonon level. Interfacing these structures requires elaborate techniques limited in tunability, flexibility, and scaling towards multi-mode systems. Here, we demonstrate a cavity optomechanical experiment using 3D-laser-written polymer membranes inside fiber Fabry-Perot cavities. Vacuum coupling rates of $g_0/2\pi \approx 30$ kHz to the fundamental megahertz mechanical mode are reached. We observe optomechanical spring tuning of the mechanical resonator frequency by tens of kilohertz exceeding its linewidth at cryogenic temperatures. The direct fiber coupling, its scaling capabilities to coupled resonator systems, and the potential implementation of dissipation dilution structures and integration of electrodes make it a promising platform for fiber-tip integrated accelerometers, optomechanically tunable multi-mode mechanical systems, and directly fiber-coupled systems for microwave to optics conversion.

Cavity optomechanical experiments have been implemented on a multitude of different platforms[1] ranging from the canonical movable end mirror of a Fabry-Perot cavity[2], over membranes in cavities[3,4], toroidal[5] and optomechanical crystal resonators[6], down to the vibrational modes of molecules in a plasmonic picocavity[7]. This platform diversification and constant improvement advanced the field during the past years leading to key achievements like ground-state cooling of mechanical resonators[8,9], optomechanical state teleportation experiments[10], efficient microwave to optical conversion[11,12], or sensing of the mechanical resonator below the standard quantum limit[13–15]. Among the current challenges in the field are the realization and addressing of multi-mode optomechanical systems and the integration of optomechanical elements for different sensing applications.

Optomechanical devices with high optical field concentration and correspondingly large optomechanical coupling are usually realized with on-chip platforms[5,6,16]. A different approach is taken by miniaturized Fabry-Perot cavities with concave mirror structures fabricated on optical fiber tips[17,18]. These fiber Fabry-Perot cavities (FFPCs) have been established as a platform for light-matter interaction during the past years, including experiments on atoms inside FFPCs for photonic

qubits or quantum networks[19,20], and realizations of optomechanical experiments[21–27]. They feature a direct fiber-coupled optical access, small cavity lengths, high optical finesse, and an open resonator volume. This allows to introduce both conventional membrane-type resonators as well as more unconventional resonators like standing waves in liquid Helium.

With its first demonstration in the 1990s[28], 3D direct laser writing (DLW) enabled the fabrication of free-formed three-dimensional polymeric structures with sub-micrometer resolution. It led to the miniaturization of on-chip optical components[29], waveguides[30], and mechanical structures[31] including 3D acoustic metamaterials[32], and has been used for mask applications[18]. Apart from large planar substrates, writing on fiber-ends has been tackled for applications in endoscopy[33] or sensing[34,35].

In this article, we demonstrate the integration of mechanical polymer membrane resonators into highly miniaturized FFPCs. We realize a miniaturized, fiber-coupled membrane-in-the-middle (MIM) experiment[3,4] with superior scaling capabilities due to the 3D DLW fabrication process both integrated into highly stable monolithic FFPCs[36] and on flat distributed Bragg reflector (DBR) substrates. We

[1]Institute of Applied Physics, University of Bonn, Bonn, Germany. [2]Institute of Physics, University of Bonn, Bonn, Germany. [3]Present address: Department of Microtechnology and Nanoscience, Chalmers University of Technology, Gothenburg, Sweden. [4]These authors contributed equally: Lukas Tenbrake, Alexander Faßbender. ✉e-mail: hannes.pfeifer@iap.uni-bonn.de

analyze the achievable optomechanical coupling strength, and demonstrate a dispersive optomechanical spring effect tuning of the mechanical resonace in the presence of a thermal optical nonlinearity exceeding the mechanical linewidth at cryogenic temperatures. To characterize the basic internal material properties no elaborate dissipation dilution structures were used in this first proof-of-principle study.

Our results pave the way for using highly flexible DLW structure fabrication for optomechanical resonators. DLW enables new realizations of high-sensitivity fiber-cavity integrated accelerometers, and mechanical multi-mode structures that are interfaced using FFPCs, including multiple membranes inside a miniaturized Fabry-Perot cavity[37] (see methods). Furthermore, extended mechanical metamaterials of membrane resonators on DBR substrates for controlling vibrations in a thin film can be realized and combination with other light-matter interfaces as 2D materials and quantum emitters can be envisaged.

## Results

### Polymer membrane in a fiber cavity

The mechanical resonator in our experiments is a drum-like polymer membrane of $1-2\,\mu m$ thickness. It is fabricated using 3D DLW (see-methods) on top of a highly reflective DBR mirror (10 ppm transmission with high reflectivity range of 750 nm to 805 nm) located either on an extended substrate or on an optical fiber-tip. A typical rectangular membrane used here spans ~ $45\,\mu m \times 60\,\mu m$ and is placed on support bars that suspend the membrane $5-10\,\mu m$ above the mirror surface (see Fig. 1a and c). The supports reduce the free membrane surface by $5\,\mu m$ on each side. More elaborate geometries like multi-membrane structures or suspensions like soft-clamping[38] can be directly realized in the 3D DLW fabrication. Finite-element simulations of the fundamental mechanical mode, see Fig. 1b and methods section, yield an expected mechanical resonance frequency of 2.1 MHz and an internal quality factor of ~ 20 due to the comparably large elastic loss tangent of the 3D DLW polymer resist material at ambient conditions. As no specific dissipation dilution is implemented in this simple design, the mechanical quality factor reflects the intrinsic $Q$ value ($Q_{int}$) of the material[39].

The membrane is integrated into an optical FFPC by approaching a fiber mirror – a fiber-tip with a concave-shaped facet and high-reflection coated surface[17,18]. The cavity length is adjusted to ~ 30 μm, but can be scanned using a piezo-electric element attached to the fiber mirror. The optical fiber leading to the approached fiber mirror serves as both input and output of the cavity. The transmission of 2000 ppm is chosen to retrieve a single-sided cavity geometry, with an approximate balance of the cavity-to-input-fiber coupling rate and the internal cavity losses. The internal cavity losses are dominated by scattering from the polymer surface that exhibits a roughness of < 5 nm (root-mean-square surface profile variation measured using atomic force microscopy) after post-fabrication polishing using an oxygen plasma ashing process. As the air-polymer interfaces of the membrane have a large transmittance, the field of the optical cavity extends through the membrane and over both open cavity domains. The optical cavity mode spectrum is characterized by scanning the cavity resonance over a probe laser tone with modulated sidebands (see methods, and [36,40]). We measure optical linewidths of $\kappa/2\pi$ of few gigahertz corresponding to finesse values of $\mathcal{F} = 1400 \pm 300$ for the case of an intensity maximum on one of the polymer-air interfaces and a minimum on the other. As scattering from the membrane surface is the dominant optical loss mechanism, the finesse reaches the empty cavity value for the case of intensity minima on both interfaces.

In order to probe the mechanical mode, we lock the cavity to the probe laser using a feedback loop on a Pound-Drever-Hall (PDH) error signal[41,42]. The thermal excitation of the membrane is transduced to cavity frequency noise visible in the noise spectrum of the calibrated PDH error signal, which is recorded using an electrical spectrum analyzer. The measured mechanical frequencies and linewidths are in good agreement with the finite element simulation with mechanical resonance frequencies between 1 and 4 MHz depending on the particular membrane geometry. The calibration of the PDH error signal slope at the lock point together with the temperature of the environment furthermore allows a quantification of the vacuum optomechanical coupling rate of the membrane modes (see methods, and [36,43]).

### Characterization of the optomechanical coupling

A linear, dispersive optomechanical coupling manifests as a shift of the optical cavity frequency $\Delta\omega_{cav}$ upon a displacement $\Delta z$ of the membrane[3,4]. In turn, the cavity field photons act on the pliable

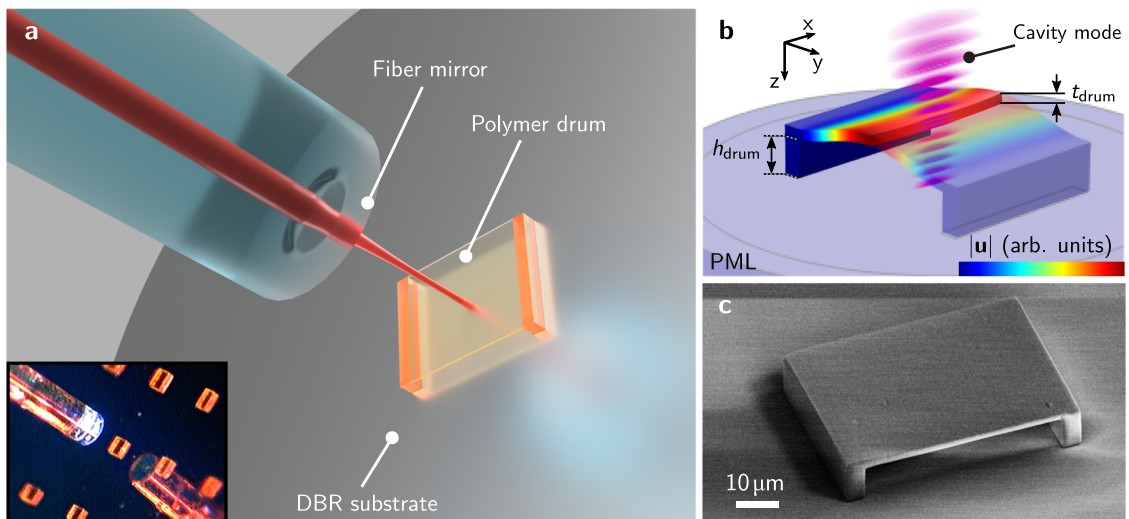

**Fig. 1 | Overview of the system.** In **a** a schematic overview of the experimental structure is shown. The polymer drum resonator is fabricated on a highly reflective DBR substrate. The fiber mirror is positioned above the polymer drum and realizes a fiber Fabry-Perot cavity with the DBR beneath enclosing the drum membrane. The inset shows a microscope picture of an approaching fiber mirror (diameter: 125 μm) to a polymer drum array. In **b** the magnitude of the displacement field **u** of the fundamental drum mode is shown as retrieved from finite element simulations. The height $h_{drum}$ of the polymer drum supports and the drum membrane thickness $t_{drum}$ determine the positions of the optical cavity mode intensity maxima and minima with respect to the membrane interfaces. In **c** an SEM micrograph of a fabricated rectangular polymer drum structure is shown.

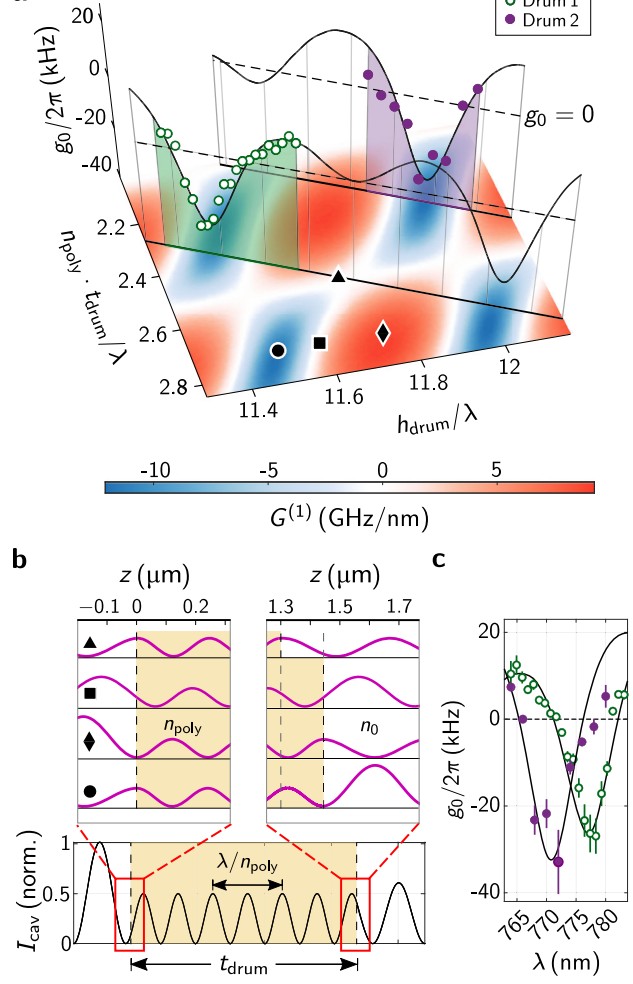

**Fig. 2 | Optomechanical coupling strengths for varying cavity-drum geometries.** In **a** the calculated $G^{(1)}(t_{drum}, h_{drum})$ are shown as a color-map, with $\{t_{drum}, h_{drum}\}$ normalized to the respective material wavelength. Measured $g_0$ for two exemplary polymer drums are highlighted in green (drum 1) and purple (drum 2) and plotted along the third plot-axis. By scanning the probe-wavelength, the measurements of the two drums trace out cuts in the map. Four cavity geometries corresponding to special coupling scenarios are highlighted: triangle and square correspond to $G^{(1)} = 0$, diamond and circle maximize $|G^{(1)}|$. In **b** (bottom) a sketch of the cavity intensity $I_{cav}$ of an exemplary cavity geometry is shown. The cavity intensity at the two polymer-air/ vacuum interfaces is shown for the four scenarios from **a** (triangle, square, diamond, circle). In **c** the measured $g_0$ from image **a** are plotted against the scanned probe laser wavelength $\lambda$ for both drum 1 (green) and drum 2 (purple). The error bars represent the combined fit and measurement uncertainty.

membrane through their radiation pressure, closing the interaction loop. The magnitude of this effect strongly depends on the geometry of the MIM system, in particular on the optical intensity on both sides of the membrane. A difference in intensity will cause a non-vanishing net radiation pressure on the membrane resulting in an optomechanical interaction. The interaction strength is given by the vacuum optomechanical coupling rate $g_0 = \frac{\partial \omega_{cav}}{\partial z} z_{zpf} := -G^{(1)} z_{zpf}$ with $z_{zpf}$ the mechanical resonator's zero-point motion amplitude. As we fix our total cavity length to ~30 μm, the relevant parameters determining the achievable coupling are the polymer drum support height $h_{drum}$ and the membrane thickness $t_{drum}$ of the drum (see Fig. 1b). Since the latter is comparable to the probe laser wavelength $\lambda$, it has a strong effect on the local cavity intensity at the membrane surfaces.

We compute the expected frequency-pull factor $G^{(1)}(t_{drum}, h_{drum})$ in our experiment using two different methods. After finding the

resonance condition of the optical cavity field (see methods), the first method considers a small shift $\Delta z$ of the membrane position. Its effect on the cavity resonance frequency as expected from Maxwell's equations is numerically extracted to determine the frequency-pull. For the second method, we employ perturbation theory for Maxwell's equations with moving material boundaries[44] (for details, see methods). For this, we use the cavity resonance condition to find the explicit form of the cavity electrical field. The resulting pull-factor (see Eq. (7)) scales linearly with the intensity difference on the two membrane-air interfaces and the polymer refractive index $n_{poly}$. Both methods yield identical results. The resulting $G^{(1)}(t_{drum}, h_{drum})$ is shown in Fig. 2a with some highlighted field configurations in b. Maximum frequency-pull factors of up to 11 GHz nm$^{-1}$ are predicted. The map features a periodic pattern of well-defined minima and maxima with a periodicity of half the (material) wavelength. The asymmetry of positive vs. negative coupling emerges from the two air domains of the cavity not having equal lengths.

We test our model by measuring $g_0$ for different probe wavelengths on two exemplary polymer geometries: $\{t_{drum\,1} \sim 1.2\,\mu m, h_{drum\,1} \sim 8.9\,\mu m\}$ and $\{t_{drum\,2} \sim 1.15\,\mu m, h_{drum\,2} \sim 9.2\,\mu m\}$. The calibrated optical cavity frequency noise spectrum $S_{\nu\nu}(f)$ is measured (see methods) and the fundamental mechanical resonance is used to extract $g_0$ at ambient conditions from its expected effect on the cavity frequency noise via[43]:

$$S_{\nu\nu}(f) = \frac{2g_0^2}{4\pi^2} \cdot \frac{2\Omega_m}{\hbar} \cdot \frac{2\Gamma k_B T}{\left(\Omega^2 - \Omega_m^2\right)^2 + \Gamma^2 \Omega^2}.$$

Here, $f = \Omega/2\pi$ denotes the noise frequency, $\Omega_m$ the mechanical resonance frequency, $\Gamma/2\pi$ the mechanical linewidth, $k_B$ Boltzmann's constant and $T$ the temperature of the polymer membrane. The results are shown in Fig. 2a and c. As the probe laser wavelength is scanned, the coupling rate changes through the shift of the cavity intensity on the polymer drum surfaces. This corresponds to the diagonal cuts in Fig. 2a across the coupling landscape for the two representative drum geometries (green – drum 1, purple – drum 2). Here, tuning the probe laser wavelength from 760 nm to 785 nm is equivalent to a few 100 nm variation of the geometry ($h_{drum}$ is chosen to be $\gtrsim 8$ μm for these geometries, such that tuning over the complete probe laser wavelength-range changes the cavity field node/anti-node positions by at least half of the (material) wavelength). We find a maximum coupling rate of $|g_0|/2\pi = 33 \pm 7$ kHz at $\lambda = 772$ nm (see Fig. 2c for drum 2). The sign of $g_0$ in Fig. 2 is inferred from the fabricated geometry parameters and the expected asymmetry of positive and negative coupling regions.

Four special cases of the intensity distribution are highlighted in Fig. 2a. Their corresponding cavity intensity distribution at the polymer membrane surfaces is shown in Fig. 2b. Vanishing optomechanical coupling is observed for equal intensities on both sides of the membrane, in the simplest case by $t_{drum}$ being a multiple of half the material wavelength. Optimal coupling is achieved, if a cavity field node (anti-node) is located at one side of the polymer membrane in combination with a corresponding anti-node (node) at the other.

Note that the quadratic frequency-pull factor is maximized for equal intensities on either side along the connection line of the linear maxima of same $t_{drum}$. This corresponds to square in Fig. 2b. It also vanishes at $t_{drum}$ being equal to a multiple of half the material wavelength triangle. The maximum expected quadratic coupling is $G^{(2)} \gtrsim 200$ GHz/nm$^2$.

## Optomechanical spring effect

We now consider dynamical effects of the intracavity photon number on the mechanical modes of the polymer membrane. As we are working in the fast-cavity/Doppler-regime, where the cavity decay rate

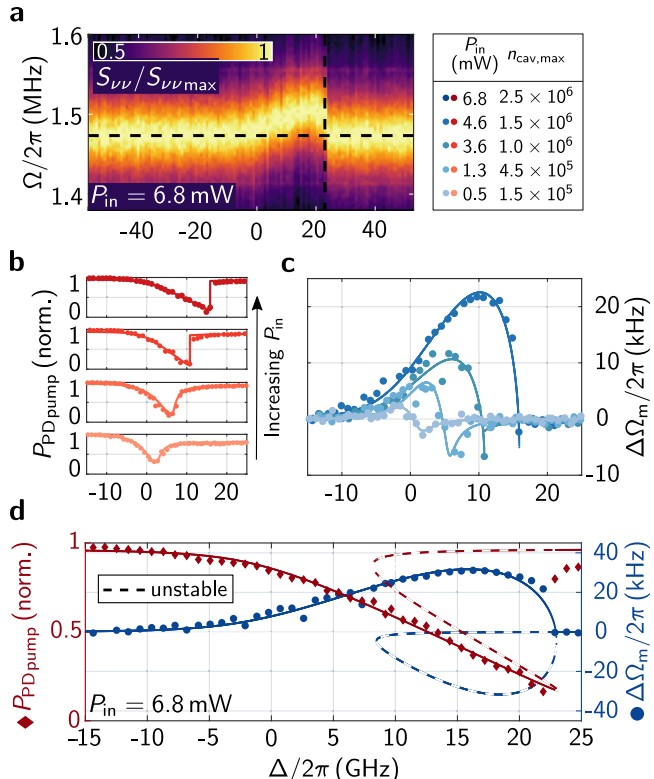

**Fig. 3 | Characterization of the optomechanical spring. a** shows the normalized frequency noise spectra $S_{\nu\nu}/S_{\nu\nu\,\text{max}}$ against cavity detuning $\Delta$ for $P_{\text{in}} = 6.8$ mW. The dashed black lines highlight the effect of the optomechanical spring on the mechanical resonance frequency $\Omega_m$. The legend in the upper right indicates the corresponding pump powers and maximum (on-resonance) cavity photon numbers for **b**-**d**. In **b** the pump-reflection signal $P_{\text{PD pump}}$ is plotted against the cavity detuning $\Delta$ for different input powers $P_{\text{in}}$. In **c** the corresponding mechanical resonance frequency shift $\Delta\Omega_m$ induced by the optomechanical spring is shown. **d** shows the normalized pump-reflection signal $P_{\text{PD pump}}$ and mechanical resonance frequency shift $\Delta\Omega_m$ induced by the optomechanical spring against cavity detuning $\Delta$ for input power $P_{\text{in}} = 6.8$ mW. The dashed lines correspond to the unstable or not-scanned solutions of Eq. (3).

$\kappa \gg \Omega_m$, the optomechanical spring effect shifts the fundamental flexural mode frequency $\Omega_m$ as

$$\Delta\Omega_m(\Delta) = g_0{}^2 n_{\text{cav}} \cdot \frac{2\Delta}{\Delta^2 + \kappa^2/4} \qquad (1)$$

with cavity detuning $\Delta$, cavity photon number $n_{\text{cav}}$ and vacuum optomechanical coupling rate $g_0$[1]. Effects on the mechanical linewidth are neglected.

To measure $\Delta\Omega_m$, we make use of a two-tone measurement scheme. The cavity is locked in a PDH feedback loop to a probe laser beam fixed at 780 nm with low optical power ($< 50\,\mu$W) (for details, see methods). $\Omega_m$ is measured from $S_{\nu\nu}(f)$ as extracted from the calibrated error signal of the feedback loop. To impose a frequency shift of the mechanical resonator, a tunable pump-laser (760 nm – 785 nm) is additionally coupled into the input-port fiber addressing another optical cavity resonance separated by one free spectral range (~5 THz) from the probe. The probe path is decoupled from the pump-laser radiation with a narrow pass-band interference filter. The pump-laser is scanned over the optical resonance modulating the detuning and the intracavity photon number $n_{\text{cav}}$. During this process, $S_{\nu\nu}(f)$ is recorded to extract $\Delta\Omega_m$. The influence of the low-power probe beam on the measurements is negligible. The measurements are repeated for a sweep of different pump input powers $P_{\text{in}}$.

Figure 3 a shows a typical measurement of $S_{\nu\nu}(f)$ of the mechanical polymer drum resonance against the cavity detuning $\Delta$ for a comparably large pump input power of $P_{\text{in}} = 6.8$ mW. At such a power level an additional photothermal nonlinearity of the optical cavity resonance shifts the resonance position during the detuning scan[45] leading to a discontinuous behavior (black dashed cross) of the signal. The photothermal nonlinearity of the optical mode is also observed in the reflection signal of the pump-laser shown in Fig. 3b. An optomechanical bistability as the origin of this behavior can be excluded as our coupling rate $g_0$ and $n_{\text{cav}}$ are too small for this effect[1]. The nonlinearity is likely related to absorption on the polymer surfaces as stronger nonlinearites are observed in geometries with intensity maxima at the polymer-air interface.

As the photothermal absorption is slow compared to the optomechanically induced dynamics, we include the shift and bistability of the optical resonance in our analysis by treating it as an additional static detuning of the cavity. The cavity detuning $\Delta$ is therefore modified to $\Delta' = \Delta - \alpha n_{\text{cav}}$ with $\alpha$ being the photothermal frequency-pull factor. To find $n_{\text{cav}}$ for a given detuning, we insert $\Delta'$ in the steady-state solution of the equation-of-motion of the complex light amplitude $\hat{a}$ as obtained from input-output formalism[46]

$$-\sqrt{\kappa_{\text{ex}}}\hat{a}_{\text{in}} = (i\Delta' - \kappa/2)\hat{a}, \qquad (2)$$

with cavity input coupling $\kappa_{\text{ex}}$ and complex input field amplitude $\hat{a}_{\text{in}}$. Squaring and averaging Eq. (2) on both sides leads to a third-order polynomial equation in the cavity photon number $n_{\text{cav}}$:

$$0 = \alpha^2 n_{\text{cav}}^3 - 2\Delta \cdot \alpha n_{\text{cav}}^2 + \left(\Delta^2 + \kappa^2/4\right)n_{\text{cav}} - \kappa_{\text{ex}}n_{\text{in}} \qquad (3)$$

with cavity photon number $n_{\text{cav}} = \langle \hat{a}^\dagger \hat{a} \rangle$ and input photon number rate $n_{\text{in}} = \langle \hat{a}_{\text{in}}^\dagger \hat{a}_{\text{in}} \rangle$. The two stable solutions of $n_{\text{cav}}$ correspond to the photon number expected from the two directions of the detuning scan, whilst the third, unstable solution is not reached[45]. As shown in Fig. 3b and d (red curve), we start the scan from $\Delta < 0$. As more and more photons are coupled into the cavity, the resonance shifts away from the approaching pump-laser following one of the stable branches for the cavity reflection signal. When the pump-scan catches up with the drifting resonance at the maximum amount of intracavity photons, any further detuning abruptly reduces the thermal frequency-drift of the resonance resulting in a sudden jump to the reflection signal baseline. Using the solutions of Eq. (3), the extracted cavity photon number $n_{\text{cav}}(\Delta)$ can be inserted in Eq. (1) with $\Delta$ being replaced by $\Delta'$. This model is then used to fit the measured data displayed in Fig. 3c and d (blue curve). The unstable solution and the not-scanned branch of the reflection signal are included as dashed lines tracing a loop shape. Towards low pump-powers, the transition to a normal dispersive lineshape of the optomechanical spring effect can be observed. The maximum optomechanical frequency shift measured here was $\Delta\Omega_m/2\pi = 31$ kHz for the maximum pump-power that we reached of $P_{\text{in}} = 6.8$ mW. The cavity finesse in this case was $\mathcal{F} = 1400 \pm 300$ and the photothermal frequency-pull factor $\alpha/2\pi = (8 \pm 2)$ kHz. Improved cavity finesse by stronger surface polishing, and higher pump-powers will allow us to increase this shift. Currently it already surpasses the mechanical linewidth at cryogenic temperatures. As the shift is on the order of the current frequency disorder of printed polymer drums it could be used to dynamically tune single drums in a multi-mode system into and out of collective resonances.

## Mechanical resonance linewidth

Under ambient atmospheric conditions, the mechanical quality factor of the polymer membrane oscillators is limited to ~20 by about equal parts through damping of the membrane motion by surrounding gas and internal losses of the polymer. In comparison, radiation of mechanical energy into the substrate is negligible due to the

impedance mismatch of sound in the polymer and the glass material below. Internal losses of the polymer are characterized by the loss tangent – the tangent of the phase between imaginary and real part of the dynamic modulus – of the material and vary strongly between different DLW resists[47]. In addition, the mechanical properties of polymers can exhibit a rich temperature dependence[48] with possible secondary glass transitions due to conformation changes of polymer chains.

To investigate the temperature dependence of the mechanical properties, we place a ferrule-based, passively stable FFPC configuration (for details, see ref. 36) in a liquid Helium continuous flow cryostat. The polymer membrane structure is fabricated on a highly reflective fiber mirror (see Fig. 4a). The sample holder, on which the FFPC is mounted, sits in an evacuated chamber and can be cooled from room temperature down to ~ 4 K. The sensor temperature is, however, only a lower bound to the local temperature of the membrane. When cooling down the membrane resonator, the polymer material stiffness and pre-strain increase, which leads to an increase of the mechanical resonance frequency from about 3 MHz to ~ 3.5 MHz (see Fig. 4b). At the same time the mechanical linewidth drops from about 150 kHz down to ~ 6 kHz corresponding to a mechanical quality factor of ~ 600 as shown in Fig. 4c. Kinks in the temperature dependence of the mechanical linewidth may indicate possible secondary glass transitions[48].

The remaining loss, decoherence or linewidth broading of the polymer membrane resonators at cryogenic temperature can be caused by various mechanisms. A possible candidate is a mechanical coupling to the conformation changes of polymer molecule chains. Also scattering of high energy phonons can appear, which would be suppressed at even lower temperatures. Due to the comparably large thickness of the membrane resonators, thermo-elastic dissipation[49,50] is expected to play another major role. Engineering the mechanical mode to reduce strain gradients and further polishing using oxygen plasma ashing to thin down the membrane will help to reduce these loss channels and can be combined with isolation and soft-clamping techniques to further increase the mechanical quality factor[38,39].

Pre-straining, together with a thinned membrane, will further help to dilute dissipation and improve the mechanical quality factor. An advantageous resource for pre-straining the material will be the usual shrinkage of the polymeric resist during polymerization. The comparably large tolerance of polymers to straining that is on par or even exceeding the limit strain of conventional materials used for micro-mechanical oscillators can enable comparably large dilution factors partly compensating the lower intrinsic mechanical quality factor. According to the manufacturer, the employed IP-S resist can show shrinkage between 2–12% under tuned fabrication conditions that were not explored here (other sintered glass-composite resists even up to 26.7%). Due to the anchoring of the structure the shrinkage in the development directly translates into strain. Assuming 10% strain, a dilution factor of ~ 280 would be reached for a 100 μm long, λ/4–thick (195 nm) structure[39,51–53]. For the cryogenic intrinsic quality factor this would result in a $Q_{mech} > 1.5 \times 10^5$ even without further engineering of the resonators. Other resists with higher intrinsic quality and even stronger shrinkage can potentially boost this prospect by another order of magnitude.

The exploration of the material properties of other 3D DLW resists under cryogenic conditions will also help to further identify low-loss materials. Moreover, 3D DLW structures can be used as blanks for subsequent material deposition, where the resist blank is later removed by oxygen plasma ashing or suit as stamp frames (e.g. from PDMS) in combination with other membranes or 2D materials like transition metal dichalcogenides.

## Discussion

We have demonstrated an FFPC-integrated optomechanical membrane-in-the-middle experiment with 3D direct laser-written

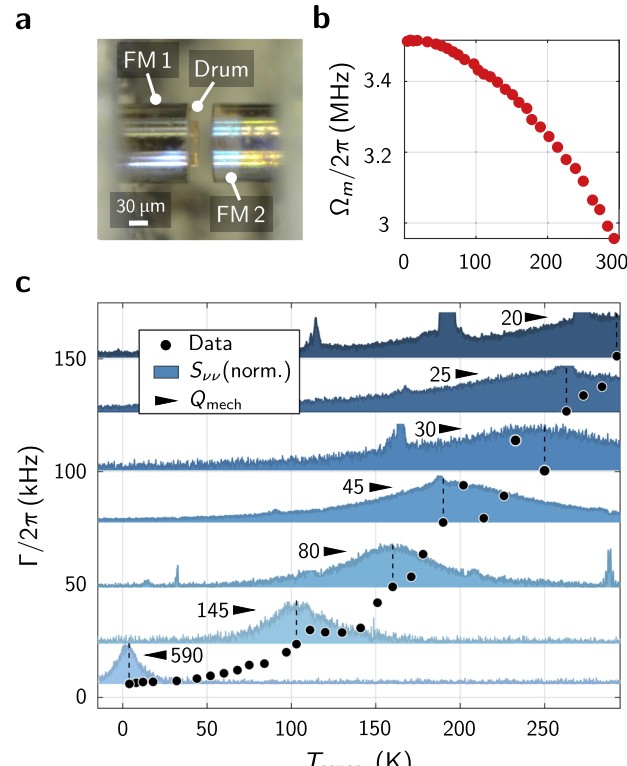

**Fig. 4 | Temperature dependence of the mechanical resonator properties.** The optomechanical cavity geometry for the experimental setup inside a liquid Helium flow cryostat uses an FFPC with two fiber mirrors (FM 1 and FM 2) inside a slotted glass ferrule that is tunable by an attached piezo[36]. The polymer drum is directly fabricated on one of the fiber mirrors as shown in **a**. Upon cool-down the polymer stiffens causing the resonance frequency to increase as shown in **b**. In **c**, the mechanical linewidth at different sensor temperatures is shown. For some exemplary measurements the quality factors and measured spectra are included.

membrane structures. Despite the relatively low membrane reflectivity, large optomechanical couplings of $g_0 \approx 30$ kHz are realized since the comparably large membrane thickness[3,21,22,26,27] of several λ/4 allows us to maximize the intensity differences on the two membrane-air interfaces. Reducing the effective membrane mass by thinning the membrane thickness down to the order of a single λ/4 thickness will further enhance the optomechanical coupling.

The unconventional polymer material stemming from the 3D DLW fabrication process results in moderate intrinsic mechanical quality factors at ambient conditions. However, a significant gain in the mechanical Q-factor is realized at cryogenic temperatures. Furthermore, the large range of specialized 3D DLW materials[47], the possible implementation of dissipation dilution techniques[39], the use of 3D DLW structures as blanks for material growth, and their capabilities as stamping frames will allow a fast advancement of the presented platform. In turn, optomechanical low-temperature experiments enable highly sensitive material characterization of DLW materials, which will contribute to the current rapid development of specialized resists.

The large flexibility of the 3D DLW fabrication allows the combination of this highly integrated platform with additional structures like electrodes[18] for electromechanical coupling, or emitters, and makes it scalable towards multi-membrane experiments. These can be both realized as membrane-stacks in a single FFPC that are considered to lead to improved values of the optomechanical coupling[37,54], or as a planar 2D mechanical metamaterial on a DBR substrate. First experiments, where we introduce stacks of two mechanical membranes in a single optical cavity, see Fig. 5, show the feasibility of this approach. In

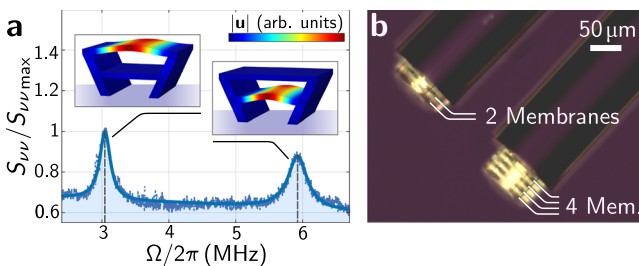

**Fig. 5 | Multi-membrane structures. a** shows the optical noise spectrum of a cavity with two mechanical membrane modes and their corresponding simulated displacement fields. The modes in this example structure are deliberately detuned by tilting the supports and thereby the size of the corresponding membrane.
**b** Showcase example of multi-mode mechanical structures realized using direct laser-writing showing stacks of multiple, freely suspended drums (here 2 and 4) that are directly fabricated on the tip of optical fibers.

contrast to other platforms[55,56] the number of layers in such stacks can easily be extended beyond two and the geometry of each membrane separately adjusted to bring multiple membranes in tune. Using DLW, mechanical device layers can also be added and combined with a plethora of other platforms for light-matter interaction. Acoustic metamaterials in such device layers would benefit from the large optomechanical spring effects allowing for tunable mechanics and optical reconfiguration of mechanical multi-mode circuits with much less elaborate tuning techniques than required in other optomechanical platforms[57]. This will enable vibration routing in 2D metamaterials, distributed sensing in multi-mode mechanical structures, and fiber-tip-integrated sensing of motion and force.

## Methods
### Sample fabrication
The polymer drum structures were fabricated using a commercial 3D lithography system (Nanoscribe Photonic Professional GT+, Nanoscribe GmbH & Co. KG, Germany) that uses two-photon-polymerization. A $63\times$ objective was used in combination with the photoresist IP-S in dip-in configuration to print on the end-facet of the single-mode fiber or DBR substrates using the system's piezo-mode. The system features a femtosecond laser centered at 780 nm that patterns and polymerizes the resist at the objective's focus. The hatching distance for the feet and frame were chosen comparably coarse, while the membrane was patterned fine (65 nm) using a laser-power of 22 % (corresponding to 11 mW) and a scanspeed of 50 μm/s. The unpolymerized resist part was removed via immersion in propylene glycol methyl ether acetate (PGMEA) (30 min) and a subsequent bath in isopropyl alcohol (30 min). In a post-development step, the structure was flood-exposed with a 6 W UV-lamp while sitting in isopropyl alcohol (10 min). Whilst this procedure can minimize structure deformation during the development, it also only results in minimal shrinkage of the resist. Under certain fabrication conditions IP-S is specified to exhibit 2–12% shrinkage. For comparison, the silica-compound based GP-Silica resist can even show up to 26.7% shrinkage in the sintering step alone. Further post-processing of the membrane was performed using oxygen plasma-ashing for surface polishing to reduce scattering losses from the interfaces. The polymer structures were fabricated on the DBR substrate or fiber mirror that constitute the higher reflective cavity mirror (10 ppm transmission). To reduce the complexity of the analysis and to focus on the fundamental material properties of the resist, the comparably simple rectangular membrane design is presented in this article. However, using this process, several different geometries were realized and much more complex structures, including for example membrane stacks as shown in Fig. 5 can be fabricated. The thinnest membrane size in such structures can be reduced below the voxel height $h_v$ (smallest voxel:

$(\oslash_v, h_v) \sim (0.2, 0.7)$ μm) by oxygen plasma-ashing. The accuracy of the placement of the membranes is given by the piezo stage positioning accuracy (10 nm in the full 300 μm × 300 μm writing field), which is considerably smaller than $\lambda/4$ as required for the membrane positioning. Possible tilts $\alpha_{tilt}$ of the structure are corrected by the automatic interface finder, but would also only play a role for tilts approaching $\alpha_{tilt} \sim (\lambda/4)/L_{mem}$ with $L_{mem}$ the length of the membranes. Detailed characterizations of such multi-membrane devices will be presented in future work.

### Experimental setup
The full experiment setup is sketched in Supplementary Fig. 1. The optical mode is characterized by measuring the reflected optical power of a fixed wavelength probe laser (probe setup red highlighted in Supplementary Fig. 1, see also [36,40]) on a photodiode ($PD_{probe}$), while the cavity length is scanned using a piezo-electric element that is glued to the optical fiber. The high-voltage drive of the piezo is generated by a $100\times$ voltage amplifier and a subsequent low pass filter (LPF) to reject high-frequency electric noise. The alignment of the fiber during the scan process is secured by a glass ferrule (131 μm bore diameter), in which the optical fiber can slide. The incoming and reflection signal are split on a polarizing beam splitter (PBS). The subsequent waveplates ($\lambda/2$ and $\lambda/4$) are used to adjust the polarization of the reflected light to be perpendicular to the polarization of the incoming laser beam causing the PBS to direct most of the reflection onto $PD_{probe}$. An electro-optic modulator (EOM) adds sidebands to the main probe laser tone. The reflection signal measured with the oscilloscope therefore features three reflection dips with a spacing of the RF-drive of the EOM that is thereby used as a frequency meter.

To characterize the optomechanical coupling strength the optical cavity is locked to the probe laser tone using a Pound-Drever-Hall (PDH, blue highlighted) or side-of-fringe (SoF) lock. The slope of the lock signal is calibrated using the frequency meter enabled by the EOM-sidebands. Using this, the measured voltage noise of $PD_{probe}$ can be converted to the optical cavity frequency noise. As the bandwidth of the feedback loop is small (~ 1 kHz) compared to the mechanical resonance frequency ($\gtrsim 1$ MHz) a bias tee is used to split the low from the high-frequency component. The DC-like part is used in the feedback loop, whilst the high-frequency part is analyzed using an electric spectrum analyzer (ESA). Mechanical mode frequency and linewidth can directly be extracted from there. The vacuum optomechanical coupling strength is then inferred by comparing the optical cavity frequency noise with the expected thermal noise of a mechanical oscillator of the measured frequency and linewidth at ambient temperature conditions[43].

In order to measure the dynamic optomechanical spring effect a second pump laser is used (highlighted in green), while the optical cavity remains tightly locked to the probe laser. The pump laser addresses a second optical cavity mode separated by one free spectral range towards higher frequencies from the probe-cavity-resonance. For variable pump laser powers, the pump frequency is scanned. The reflection signal of the pump is recorded on a second photodiode ($PD_{pump}$), while the locked probe laser error signal is used for the characterization of the mechanical mode frequency.

### Optical properties of the membrane-cavity system
The FFPC designs used in this work are based on either a hemi-cavity design with a fiber mirror that can scan structures on a flat, highly reflective substrate inspired by[58], or a passively stable, monolithic FFPC realization with two fiber mirrors inside a glass ferrule as demonstrated in[36].

We use a standard $CO_2$ laser ablation system to machine spherical-like depressions onto the center of single-mode optical fibers[17]. The resulting concave indentation on our fiber end-facets features typical radii of curvature of ~ 200 μm with usable spherical diameters

of $> 40\,\mu m$. The prepared fiber tips are coated with alternating layers of $Ta_2O_5$ and $SiO_2$, resulting in highly reflective fiber end-facets for wavelengths ranging from $750 - 800\,nm$ at AOI = 0°. These fiber mirrors make up the cavity geometry and are coated for high transmission (2000 ppm) to be used as in-coupling fiber mirrors to the cavity and low transmission (10 ppm) blank mirrors for interfacing the direct laser-written membranes. The empty cavity finesse of ~2800 (no polymer membrane inside cavity volume) is primarily determined by the transmission losses of the cavity mirrors.

The optical losses induced by the polymer membrane at lower input laser powers ($< 500\,\mu W$) are dominated by surface scattering effects at the polymer-air interfaces. Depending on the cavity field distribution, optical scattering losses induced by the polymer membrane can range from ~ 5000 ppm with cavity intensity maxima on both polymer-air interfaces to almost fully recovering the empty cavity optical quality in the case of intensity minima on both interfaces. The scattering from the polymer surfaces can be strongly reduced by polishing through oxygen plasma (Plasma cleaner system: Zepto-BR-200-PCCE), which smoothens the surface as shown in Supplementary Fig. 2. As the plasma removes material from the surface, this technique can also be used to reduce the thickness of the membrane. Another possibility to in the future enhance the optical properties is to combine the fabrication with an atomic layer deposition-based uniform growth that can flatten the surface even further.

At higher input laser powers ($> 500\,\mu W$), additional photothermal nonlinearities of the optical cavity resonance become relevant. Since we observe stronger nonlinearities in cavity geometries with intensity maxima at the polymer-air interfaces, these effects can most likely be attributed to surface absorption as for example induced by not-passivated bonds of the polymer. The additional absorption will then lead to heating and mechanical expansion of the membrane, causing the additional static photothermal detuning as observed in the experiments.

## Coupling calculations

To compute the expected optomechanical frequency-pull factor $G^{(1)}(t_{drum}, h_{drum})$ for a particular drum thickness and support height, we use two methods as described in section II. For both methods, we first numerically find the set of $\{l_{drum}, t_{drum}, h_{drum}\}$ that fulfills the resonance condition of the optical cavity with the probe light at $\lambda = 2\pi/k_0$. Here, $h_{drum}$ is the polymer drum support height, $t_{drum}$ the polymer membrane thickness, and $l_{drum} = L_{cav} - t_{drum} - h_{drum}$, where, $L_{cav}$ denotes the separation between the two cavity mirrors (for an overview see Fig. 1b). For each pair $\{t_{drum}, h_{drum}\}$, a $l_{drum}$ at $L_{cav}$ - 30 µm can be chosen to match the resonance condition $R(l_{drum}, t_{drum}, h_{drum}) = 0$, with (suppressing subscripts):

$$R(l,h,t) = A_+(l) \cdot e^{-in_{poly}k_0 t} - A_-(h) \cdot e^{in_{poly}k_0 t} \qquad (4)$$

and with coefficients $A_\pm(z)$:

$$A_\pm(z) = \frac{1 \mp in_{poly} \cdot \tan(k_0 z)}{1 \pm in_{poly} \cdot \tan(k_0 z)}.$$

In the picture used here, the cavity consists of a loss-less dielectric membrane with refractive index $n_{poly}$. It is placed between two mirrors with perfectly conducting surfaces, enforcing field nodes at their positions (the penetration depth of the DBR will in reality lead to a slightly modified resonance cavity length). As the Rayleigh length $z_R \gg L_{cav}$, the Gaussian-beam properties of the cavity field are neglected and a simple standing-wave ansatz is made. There, the tangential component of the electric field at both mirror surfaces needs to vanish and the tangential components of both the electric- and magnetic fields need to be continuous at the dielectric interfaces[59]. Applying these conditions to the cavity field, we numerically find the condition

that fixes the $\{l_{drum}, t_{drum}, h_{drum}\}$ - triplet. This is now used to extract the optomechanical frequency-pull factor $G^{(1)}(t_{drum}, h_{drum})$:

In method 1, we use Eq. (4) by applying a small shift to the membrane, altering the cavity geometry to $l_{drum} \to l_{drum} + \Delta z$, $h_{drum} \to h_{drum} - \Delta z$ and $k_0 \to k_0 + \Delta k$. This simulates the effect of the linear displacement of the membrane center region on the cavity resonance condition (for this mathematical treatment, $h_{drum}$ does not directly correspond to the polymer drum support height, but more accurately to the distance between the displaced membrane at its center and the subjacent mirror. The mode width of the optical mode is sufficiently smaller than the effective membrane radius to ensure that the optical mode only overlaps with an almost constant displacement of the drum membrane). Inserting the shifted geometry into Eq. (4) allows to numerically extract a $\Delta k$ that again fulfills Eq. (4) and thereby reflects the shift on the resonance frequency $\Delta\omega_{cav}$. The ratio between the frequency shift and the small displacement allows us to numerically determine the frequency-pull $G^{(1)}(t_{drum}, h_{drum})$. Aside from the linear frequency-pull factor, higher orders can also be numerically evaluated. E.g. for the quadratic pull-factor values of $\gtrsim 200$ $GHz\,nm^{-2}$ are expected.

For method 2, we use the optical field that fulfills the resonant cavity condition and apply perturbation theory to find the shift of the optical resonance associated with a shift of the dielectric boundaries of our geometry as derived by Johnson et al. in [44]. Here, Maxwell's equations are written as an eigenproblem of the electric cavity field $|E\rangle$ (for convenience, the basis-independent representation of the electric field as bra-ket vectors is utilized, with inner product $\langle E|E'\rangle \equiv \int \boldsymbol{E}^* \cdot \boldsymbol{E}'\, dV$) with eigenfrequency $\omega_{cav}$ given by the well-known source-free wave equation with overall cavity permittivity $\epsilon(z)$:

$$\nabla^2 |E\rangle = \left(\frac{\omega_{cav}}{c}\right)^2 \epsilon(z)|E\rangle. \qquad (5)$$

Due to the vibrational motion of the drum membrane, the effective dielectric permittivity of the cavity geometry experiences a local shift $\delta\epsilon$ due to a perturbative shift in position of the drum membrane $\delta z$. We expand $|E\rangle$, $\omega_{cav}$ to first-order in $dz$. Plugging these expressions back into Eq. (5) and neglecting terms of $\mathcal{O}(dz^2)$ leads to the first-order correction of the resonator frequency in differential form[44]:

$$G^{(1)} = \frac{d\omega_{cav}^{(1)}}{dz} = -\frac{\omega_{cav}^{(0)}}{2} \frac{\left\langle E^{(0)} \left| \frac{d\epsilon}{dz} \right| E^{(0)} \right\rangle}{\left\langle E^{(0)} | \epsilon | E^{(0)} \right\rangle} \qquad (6)$$

We then make use of the explicit parametrization of the cavity permittivity

$$\epsilon(z) = \epsilon_2 + \Delta\epsilon\big(\Theta(z - z_0) - \Theta(z - (z_0 + t_{drum}))\big),$$

with $\Delta\epsilon = \epsilon_1 - \epsilon_2$ ($\epsilon_1$: polymer, $\epsilon_2$: air/vacuum) and the Heaviside step function $\Theta(z)$. The first drum surface $S_L$ is located at $z_0$ and the second drum surface $S_R$ at $z_0 + t_{drum}$ following the coordinate conventions of Fig. 2b.

Inserting this expression back into Eq. (6), we arrive at the explicit form of the optomechanical frequency-pull $G^{(1)}$:

$$G^{(1)} = \frac{d\omega_{cav}}{dz} = \frac{\omega_{cav}^{(0)}}{2} \frac{\int_{S_R} \Delta\epsilon \left| \boldsymbol{E}_\parallel^{(0)} \right|^2 dS - \int_{S_L} \Delta\epsilon \left| \boldsymbol{E}_\parallel^{(0)} \right|^2 dS}{\int_V \epsilon(z) \left| \boldsymbol{E}_\parallel^{(0)} \right|^2 dV}, \qquad (7)$$

with the total resonator volume $V$ and parallel electric field component $\boldsymbol{E}_\parallel^{(0)}$ (explicit form given by Eq. (4)). As detailed in section II, Eq. (7) allows us to read-off the frequency-pull $G^{(1)}(t_{drum}, h_{drum})$ for a specific cavity geometry by considering the cavity field distribution on both surfaces of the polymer drum. It also provides the physical

interpretation of the coupling to be caused by differing field intensities on the membrane-air interfaces. The coupling landscape retrieved from method 1 and method 2 agrees for all tested geometries. For our cavity geometry of interest, the results are shown in Fig. 2a.

### Finite element simulations

We use COMSOL Multiphysics®[60] to perform finite element simulations of the mechanical resonator structures. The simulated geometry consists of the polymer membrane (IP-S: Dynamic modulus $E = (5.33 + 0.26i)$ GPa, Poisson's ratio $v = 0.3$, density $\rho = 1.15$ kg dm$^{-3}$ [47]) with a silica substrate below. An outer shell of the substrate is defined as a perfectly matched layer (PML) to implement radiation losses into the substrate (see Fig. 1b). Whilst these radiation losses are not the dominating loss mechanism at room temperature, they are strongly dependent on the polymer geometry and can be reduced to limiting quality factors of $\gg 10^6$ for optimized support dimensions and with included isolation cuts. By reducing the internal polymer losses, e.g. through other resist materials, high-quality factor mechanical resonators would be feasible. Aside from the quality factor and resonance frequency, we use the simulation to extract the effective mass of the fundamental flexural mode (on the order of ~ 2.4 ng depending on the specific geometry) and the corresponding zero point motion (~ 3.2 fm)[61].

### Data availability

The raw data underlying the results presented in this paper is freely accessible in the open-access Zenodo database under https://doi.org/10.5281/zenodo.10204568.

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

## Acknowledgements

The authors would like to thank Prof. Dieter Meschede for supporting the first experiments on this project. The authors acknowledge funding by the Deutsche Forschungsgemeinschaft (DFG, German Research Foundation) under Germany's Excellence Strategy - Cluster of Excellence Matter and Light for Quantum Computing (ML4Q) EXC 2004/1 - 390534769 as well as funding from the Bundesministerium für Bildung und Forschung (BMBF, Federal Ministry for Education and Research) - project FaResQ. S.H. furthermore acknowledges funding from the European Union's Horizon 2020 program under the ERC consolidator grant RYD-QNLO (Grant No. 771417).

## Author contributions

H.P., L.T., A.F., and S.L. came up with the concept and planned the experiments. A.F., and L.T. performed the fabrication, and L.T. conducted the optical measurements. L.T., A.F., and H.P. set up the numerical simulations and analyzed the data. H.P, S.H., and S.L. supervised the research and were involved in the funding acquisition. All authors contributed to the writing of the manuscript.

## Funding

## Competing interests

The authors declare no competing interests.
