## [Peer Review File · Nature Communications]

DLW optomechanical membranes in FFPCsEditorial Note: This manuscript has been previously reviewed at another journal that is not operating a transparent peer review scheme. This document only contains reviewer comments and rebuttal letters for versions considered at *Nature Communications*.

REVIEWER COMMENTS

Reviewer #1 (Remarks to the Author):

After reading the revised manuscript after the second round and the reply to the referees' comments, I am generally not convinced that this work is at the level of *Nature Communications* in terms of impact. It is certainly innovative and of high quality, but polymer membranes are not (yet?) very good candidates for most of the optomechanical applications envisaged by the authors in terms of both optical and mechanical properties. I share in particular the same opinion as Reviewers 1 and 4 on the mechanical Q issues. It is praiseworthy to investigate how to achieve high Qs with this type of membranes, but at this point the prospects are too speculative. I agree with the authors that the fabrication of multielement arrays is the main demonstrated advantage, but a specific application showcasing this advantage is not demonstrated. So, unless either polymer membranes with competitive optical/mechanical properties can be fabricated or the multielement nature of the structure can be exploited decisively in an application, I would not recommend publication of this work in *Nature Communications*, but rather in a more specialized journal.

Reviewer #2 (Remarks to the Author):

I believe this is a very interesting paper that certainly deserves publication in *Nat. Comm.* I've heard for a long time people talking about using such laser-written systems for optomechanics experiments but this is the very first convincing system I have seen.

I understand there were a number of discussions between the authors and the numerous referees and I believe all issues have been convincingly addressed by the authors.

To summarize the main problems spotted:

- Does the simple demonstration of such a system deserve publication in *Nat. Comm.* or do the authors need to perform a specific state-of-the-art experiment (cooling, force sensing...) to motivate the publication (as evidenced by "...the reported system remains at insufficient values, far from other systems...")?

I would go for the former: a promising system (even with all its current limitations, especially in terms of Q) like this certainly deserves publication.

I understand though this is not the point of view of some referees.

- Are the current optical limitations an issue?

I do not think so. There was much (incremental) progress demonstrated "... the limitations due to surface roughness are now discussed with greater care...") and I think even higher quality is certainly

possible. Also, the discussion

Also, the discussion of the optical properties (including the curved fiber mirror) is very good I believe.

- Is the Q factor issue a real issue?

I understand there has been an intense conversation between former referees and the authors about this.

I think even though the currently attained values are still modest, there is a roadmap and a reasonable hope to reach

much higher values. I disagree with reviewer #1 when he states "... the limited Q factor... is now addressed in a highly speculative manner...".

I believe on the contrary that the authors have put a real effort into providing the readers with relevant information and that their reply to the reviewer is satisfactory.

Just one point which does not seem satisfactory to me right now is the (too short) discussion about the possibility to

use multiple structures. Apart from the possibility to create such structures, I think the discussion requires details about

how the authors think they can tilt and position them with the required accuracy.

To conclude, I feel everyone agrees to believe these new optomechanical structures are promising and certainly worth pursuing the effort.

The question is whether or not the current results deserve publication. I certainly think so.

Reviewer #3 (Remarks to the Author):

The criticisms of the previous review round have been all addressed with care.

- The central point of limited Q and the potential to improve it via dissipation dilution has been now clarified to a level that is satisfactory. However, almost nothing of the arguments was incorporated into the manuscript, and as evidenced by Reviewers 1 and 4 posing the same question, one can expect that many readers would be interested in more details on the prospects for dissipation dilution. The authors should thus consider to add more detail on this, e.g. in one of the appendices.

- The criticism of Reviewer 4, that at least one novel, competitive aspect should be presented, is addressed. A fabricated multi-membrane structure is shown, a clear example of the potential of 3D laser writing. To make this convincing, it should be stated whether the membranes were already fully free-standing and thus in principle functional for optomechanical experiments.

Overall 3D laser printing is introduced as a novel method for realizing optomechanical structures with possible advantages, even though the potential in terms of mechanical and optical quality factors remains to be shown. I think the work will be interesting to a broader readership, and I recommend publication in *Nature Communications*.

First of all, we want to express our thanks to the reviewers for their careful review of our revised manuscript. We are grateful that all acknowledge the innovation and quality of our experiments and we believe that their feedback gave a strong contribution to a further improvement of the manuscript.

In the following, we want to address the reviewer comments with the respective reviewers' points marked in **black** while our responses are in **blue**. Changes of the manuscript that address the respective point are briefly summarized in **purple**. In the revised manuscript, changes within this round also appear in **purple**, and the revisions of previous rounds remain in **red** (round 1) and **dark red** (round 2).

On the last pages of this letter an overview list of all implemented changes is given.

With best regards,
the authors

Reviewer #1:

“After reading the revised manuscript after the second round and the reply to the referees' comments, I am generally not convinced that this work is at the level of Nature Communications in terms of impact. It is certainly innovative and of high quality, but polymer membranes are not (yet?) very good candidates for most of the optomechanical applications envisaged by the authors in terms of both optical and mechanical properties. I share in particular the same opinion as Reviewers 1 and 4 on the mechanical Q issues. It is praiseworthy to investigate how to achieve high Qs with this type of membranes, but at this point the prospects are too speculative. ...”

[Authors' response]: We are pleased that the reviewer finds the presented research innovative and of high quality. We also agree that realizing polymer membrane resonators with high quality factor will be a challenging task. However, we think that there is enough evidence that this a viable route and that the fabrication flexibility gained through the 3D direct laser writing approach will allow to fabricate previously unfeasible mechanical resonator structures. To strengthen our reasoning on why high quality factors can be achieved, also following the advice of the other two reviewers, we extended the discussion on dissipation dilution in the subsection “*Mechanical resonance linewidth*” and included some arguments from the previous exchange with the reviewers.

Manuscript changes:

- added/ rephrased paragraph , lines 338-356:
“... *An advantageous resource for pre-straining the material will be the usual shrinkage of the polymeric resist during polymerization. The comparably large tolerance of polymers to straining that is on par or even exceeding the limit strain of conventional materials used for micro-mechanical oscillators can enable comparably large dilution factors partly compensating the lower intrinsic mechanical quality factor. According to the manufacturer, the employed IP-S resist can show shrinkage between 2-12 % under tuned fabrication conditions that were not explored here (other sintered glass-composite resists even up to 26.7 %). Due*”

to the anchoring of the structure the shrinkage in the development directly translates into strain. Assuming 10 % strain, a dilution factor of ~ 280 would be reached for a 100 μm long, λ/4-thick (195 nm) structure [39, 51–53]. For the cryogenic intrinsic quality factor this would result in a $Q_{\text{mech}} > 1.5 \times 10^5$ even without further engineering of the resonators. Other resistors with higher intrinsic quality and even stronger shrinkage can potentially boost this prospect by another order of magnitude. ...”

“... I agree with the authors that the fabrication of multielement arrays is the main demonstrated advantage, but a specific application showcasing this advantage is not demonstrated. So, unless either polymer membranes with competitive optical/mechanical properties can be fabricated or the multielement nature of the structure can be exploited decisively in an application, I would not recommend publication of this work in Nature Communications, but rather in a more specialized journal.”

[Authors' response]: The main focus of this manuscript is to showcase the novel fabrication opportunities that 3D direct laser writing can provide, to demonstrate its applicability for realizing optomechanical resonators, and to discuss the physics and peculiarities that come with this approach. Multi-element structures will be a main focus of the continuing research with this platform, but a fully fledged discussion of a multi-mode structure would go beyond the scope and possible length of this article. To nevertheless provide a decisive argument that this platform is suited for optically interfaced multi-mode mechanical structures, we now include a measurement of a double membrane, where two mechanical modes are simultaneously interfaced, in the discussion part to provide an even clearer outlook. We also shifted a part of the methods section that showcases the possibility to fabricate multi-mode mechanical structures to the discussion. We hope that the reviewer can agree that this additional measurement and discussion provide a convincing and decisive argument that multi-mode mechanical structures are not only speculation, but a clear and close application of this technique.

Manuscript changes:

- extended and shifted Figure 5:
 - added subfigure part a and modified caption
“Figure 5. a. shows the optical noise spectrum of a cavity with two mechanical membrane modes and their corresponding simulated displacement fields. The modes in this example structure are deliberately detuned by tilting the supports and thereby the size of the corresponding membrane. b Showcase example of multi-mode mechanical structures realized using direct laser-writing. Stacks of multiple, freely suspended drums (here 2 and 4) that are directly fabricated on the tip of an optical fiber.”
 - moved from *Methods* subsection *Sample fabrication* to the *Discussion* section
- in lines 395–403, extension of paragraph in the *Discussion* section:
“... These can be both realized as membrane-stacks in a single FFPC that are considered to lead to improved values of the optomechanical coupling [37, 54], or as a planar 2D mechanical metamaterial on a DBR substrate. First experiments, where we introduce stacks of two mechanical membranes in a single optical cavity, see Fig. 5, show the feasibility of this approach. In contrast to other platforms [55, 56] the number of layers in such stacks can easily be extended beyond two and the geometry of each membrane separately adjusted to bring multiple membranes in tune”

Reviewer #2:

"I believe this is a very interesting paper that certainly deserves publication in Nat. Comm.

I've heard for a long time people talking about using such laser-written systems for optomechanics experiments but this is the very first convincing system I have seen. ..."

[Authors' response]: We are pleased that the reviewer acknowledges that we present a convincing system for direct laser-written optomechanical structures.

"... I understand there were a number of discussions between the authors and the numerous referees and I believe all issues have been convincingly addressed by the authors.

To summarize the main problems spotted:

- Does the simple demonstration of such a system deserve publication in Nat. Comm. or do the authors need to perform a specific state-of-the-art experiment (cooling, force sensing...) to motivate the publication (as evidenced by "...the reported system remains at insufficient values, far from other systems...")?

I would go for the former: a promising system (even with all its current limitations, especially in terms of Q) like this certainly deserves publication.

I understand though this is not the point of view of some referees. ..."

[Authors' response]: We agree with the reviewer that a thorough characterization of a promising experimental platform is a valuable contribution in its own right and a necessity to prepare the grounds for further research.

"... - Are the current optical limitations an issue?

I do not think so. There was much (incremental) progress demonstrated ("... the limitations due to surface roughness are now discussed with greater care...") and I think even higher quality is certainly possible. Also, the discussion

Also, the discussion of the optical properties (including the curved fiber mirror) is very good I believe. ..."

[Authors' response]: We thank the reviewer for his appreciation and agree that higher optical quality is indeed achievable when we further optimize the fabrication.

"... - Is the Q factor issue a real issue?

I understand there has been an intense conversation between former referees and the authors about this.

I think even though the currently attained values are still modest, there is a roadmap and a reasonable hope to reach

much higher values. I disagree with reviewer #1 when he states "... the limited Q factor... is now addressed in a highly speculative manner...".

I believe on the contrary that the authors have put a real effort into providing the readers with relevant information and that their reply to the reviewer is satisfactory.
...

[Authors' response]: We are very pleased that the reviewer follows our argument. As noted for the previous reviewer, we now have also included a bigger fraction of the answers to reviewer questions in the main text in the subsection "*Mechanical resonance linewidth*" to provide an easier access to these arguments for the readers.

Manuscript changes:

- added/ rephrased paragraph, lines 338-356:

"... An advantageous resource for pre-straining the material will be the usual shrinkage of the polymeric resist during polymerization. The comparably large tolerance of polymers to straining that is on par or even exceeding the limit strain of conventional materials used for micro-mechanical oscillators can enable comparably large dilution factors partly compensating the lower intrinsic mechanical quality factor. According to the manufacturer, the employed IP-S resist can show shrinkage between 2-12 % under tuned fabrication conditions that were not explored here (other sintered glass-composite resists even up to 26.7 %). Due to the anchoring of the structure the shrinkage in the development directly translates into strain. Assuming 10 % strain, a dilution factor of ~ 280 would be reached for a 100 μm long, λ/4-thick (195 nm) structure [39, 51–53]. For the cryogenic intrinsic quality factor this would result in a $Q_{mech} > 1.5 \times 10^5$ even without further engineering of the resonators. Other resists with higher intrinsic quality and even stronger shrinkage can potentially boost this prospect by another order of magnitude. ..."

"... Just one point which does not seem satisfactory to me right now is the (too short) discussion about the possibility to use multiple structures. Apart from the possibility to create such structures, I think the discussion requires details about how the authors think they can tilt and position them with the required accuracy. ..."

[Authors' response]: The limits for the positioning and tilt of multilayer structures are set by the specifications of the 3D direct laser writing machine. We have now summarized those for the employed machine from Nanoscribe in the methods sections "*Sample fabrication*". The experimentally relevant physical size to compare those with is the wavelength of the optical cavity. As the accuracy of the Nanoscribe is exceeding the requirements by more than an order of magnitude, we do not expect significant limitations in the structure fabrication due to this.

Manuscript changes:

- added accuracy information, lines 449-460:

"... The thinnest membrane size in such structures can be reduced below the voxel height h_v (smallest voxel: $(\varnothing_v, h_v) \sim (0.2, 0.7) \mu\text{m}$) by oxygen plasma-ashing. The accuracy of the placement of the membranes is given by the piezo stage positioning accuracy (10 nm in the full $300 \mu\text{m} \times 300 \mu\text{m}$ writing field), which is considerably smaller than λ/4 as required for the membrane positioning. Possible tilts α_{tilt} of the structure are corrected by the automatic interface finder, but would also only play a role for tilts approaching $\alpha_{\text{tilt}} \sim (\lambda/4)L_{\text{mem}}$ with L_{mem} the length of the membranes. Detailed characterizations of such multi-membrane devices will be presented in future work. ..."

“... To conclude, I feel everyone agrees to believe these new optomechanical structures are promising and certainly worth pursuing the effort. The question is whether or not the current results deserve publication. I certainly think so.”

[Authors' response]: We thank the reviewer for his positive evaluation.

Reviewer #3:

“The criticisms of the previous review round have been all addressed with care.
- The central point of limited Q and the potential to improve it via dissipation dilution has been now clarified to a level that is satisfactory. However, almost nothing of the arguments was incorporated into the manuscript, and as evidenced by Reviewers 1 and 4 posing the same question, one can expect that many readers would be interested in more details on the prospects for dissipation dilution. The authors should thus consider to add more detail on this, e.g. in one of the appendices. ...”

[Authors' response]: We are pleased that the reviewer acknowledges that we well-addressed the reviewer's questions. We also agree that more of the discussion on the Q-factor limitation should appear within the manuscript. We therefore extended the discussion on dissipation dilution in the subsection “*Mechanical resonance linewidth*” and included some arguments from the previous exchange with the reviewers.

Manuscript changes:

- added/ rephrased paragraph , lines 338-356:

“... An advantageous resource for pre-straining the material will be the usual shrinkage of the polymeric resist during polymerization. The comparably large tolerance of polymers to straining that is on par or even exceeding the limit strain of conventional materials used for micro-mechanical oscillators can enable comparably large dilution factors partly compensating the lower intrinsic mechanical quality factor. According to the manufacturer, the employed IP-S resist can show shrinkage between 2-12 % under tuned fabrication conditions that were not explored here (other sintered glass-composite resists even up to 26.7 %). Due to the anchoring of the structure the shrinkage in the development directly translates into strain. Assuming 10 % strain, a dilution factor of ~ 280 would be reached for a 100 μm long, λ/4-thick (195 nm) structure [39, 51–53]. For the cryogenic intrinsic quality factor this would result in a $Q_{mech} > 1.5 \times 10^5$ even without further engineering of the resonators. Other resists with higher intrinsic quality and even stronger shrinkage can potentially boost this prospect by another order of magnitude. ...”

“ ... - The criticism of Reviewer 4, that at least one novel, competitive aspect should be presented, is addressed. A fabricated multi-membrane structure is shown, a clear example of the potential of 3D laser writing. To make this convincing, it should be stated whether the membranes were already fully free-standing and thus in principle functional for optomechanical experiments. ...”

[Authors' response]: We are happy that the reviewer shares our assessment regarding the demonstration of a novel, competitive aspect. Indeed the shown structures are free standing and in principle fully functional. To more decisively demonstrate their applicability for multi-mode optomechanics, we now include also a measurement of a simple mechanical two-mode structure. This also directly proves the release and function of the multi-membrane structures.

Manuscript changes:

- extended and shifted Figure 5:
 - added subfigure part **a** and modified caption
 - “Figure 5. a. shows the optical noise spectrum of a cavity with two mechanical membrane modes and their corresponding simulated displacement fields. The modes in this example structure are deliberately detuned by tilting the supports and thereby the size of the corresponding membrane. b Showcase example of multi-mode mechanical structures realized using direct laser-writing. Stacks of multiple, freely suspended drums (here 2 and 4) that are directly fabricated on the tip of an optical fiber.”*
 - moved from *Methods* subsection *Sample fabrication* to the *Discussion* section
- in lines 395-403, extension of paragraph in the *Discussion* section:
 - “... These can be both realized as membrane-stacks in a single FFPC that are considered to lead to improved values of the optomechanical coupling [37, 54], or as a planar 2D mechanical metamaterial on a DBR substrate. First experiments, where we introduce stacks of two mechanical membranes in a single optical cavity, see Fig. 5, show the feasibility of this approach. In contrast to other platforms [55, 56] the number of layers in such stacks can easily be extended beyond two and the geometry of each membrane separately adjusted to bring multiple membranes in tune . . .”*

“... Overall 3D laser printing is introduced as a novel method for realizing optomechanical structures with possible advantages, even though the potential in terms of mechanical and optical quality factors remains to be shown. I think the work will be interesting to a broader readership, and I recommend publication in Nature Communications.”

[Authors' response]: We agree with the assessment of the reviewer and want to thank the reviewer for his positive evaluation.

Overview list of text changes:

Changes of this round are also highlighted in the revised manuscript in **purple**.

- in line 333, change to plural formulation:
“... *these loss channels* ...”
- in lines 336-356, formulation of a separate paragraph on the prospect of realizing dissipation dilution with polymer structures:
“... ~~Pre-straining, as induced by the usual shrinkage of the polymeric resist during polymerization (see methods), together with a thinned membrane, will further help to dilute dissipation and improve the mechanical quality factor by several orders of magnitude. Anticipating similar improvements of $Q_{\text{mech}} / Q_{\text{int}} \sim 10^4$ by dissipation dilution, as regularly achieved on other platforms, would allow for mechanical quality factors exceeding one million using the same DLW resist material.~~ An advantageous resource for pre-straining the material will be the usual shrinkage of the polymeric resist during polymerization. The comparably large tolerance of polymers to straining that is on par or even exceeding the limit strain of conventional materials used for micro-mechanical oscillators can enable comparably large dilution factors partly compensating the lower intrinsic mechanical quality factor. According to the manufacturer, the employed IP-S resist can show shrinkage between 2-12 % under tuned fabrication conditions that were not explored here (other sintered glass-composite resists even up to 26.7 %). Due to the anchoring of the structure the shrinkage in the development directly translates into strain. Assuming 10 % strain, a dilution factor of ~ 280 would be reached for a 100 μm long, $\lambda/4$ -thick (195 nm) structure [39, 51–53]. For the cryogenic intrinsic quality factor this would result in a $Q_{\text{mech}} > 1.5 \times 10^5$ even without further engineering of the resonators. Other resists with higher intrinsic quality and even stronger shrinkage can potentially boost this prospect by another order of magnitude. ...”
- in lines 395-403, extension of paragraph in the *Discussion* section:
“... These can be both realized as membrane-stacks in a single FFPC that are considered to lead to improved values of the optomechanical coupling [37, 54], or as a planar 2D mechanical metamaterial on a DBR substrate. First experiments, where we introduce stacks of two mechanical membranes in a single optical cavity, see Fig. 5, show the feasibility of this approach. In contrast to other platforms [55, 56] the number of layers in such stacks can easily be extended beyond two and the geometry of each membrane separately adjusted to bring multiple membranes in tune”
- extended and shifted Figure 5:
→ added subfigure part **a** and modified caption
“Figure 5. **a.** shows the optical noise spectrum of a cavity with two mechanical membrane modes and their corresponding simulated displacement fields. The modes in this example structure are deliberately detuned by tilting the supports and thereby the size of the corresponding membrane. **b** Showcase example of multi-mode mechanical structures realized using direct laser-writing. Stacks of multiple, freely suspended drums (here 2 and 4) that are directly fabricated on the tip of an optical fiber.”
→ moved from *Methods* subsection *Sample fabrication* to the *Discussion* section
- in lines 433-438 and 449-460, restructuring of the *Methods - Sample fabrication* subsection
shifted information (433-438):
“... Whilst this procedure can minimize structure deformation during the development, it also

only results in minimal shrinkage of the resist. Under certain fabrication conditions IP-S is specified to exhibit 2-12 % shrinkage. For comparison, the silica-compound based GP-Silica resist can even show up to 26.7% shrinkage in the sintering step alone. ... “

removed from end of the section:

~~“... Shrinkage of photoresists during polymerization or sintering (e.g. IP-S: 2-12 % or GP-Silica: 26.7 % during sintering) can furthermore be exploited to implement dissipation dilution techniques for pre-strained materials [39,52] that for realistically achievable geometries can boost the mechanical quality factor by several orders of magnitude. Detailed characterizations of such devices will be presented in future work. ...”~~

added accuracy information (449-460):

“... The thinnest membrane size in such structures can be reduced below the voxel height h_v (smallest voxel: $(\varnothing_v, h_v) \sim (0.2, 0.7) \mu\text{m}$) by oxygen plasma-ashing. The accuracy of the placement of the membranes is given by the piezo stage positioning accuracy (10 nm in the full $300 \mu\text{m} \times 300 \mu\text{m}$ writing field), which is considerably smaller than $\lambda/4$ as required for the membrane positioning. Possible tilts α_{tilt} of the structure are corrected by the automatic interface finder, but would also only play a role for tilts approaching $\alpha_{\text{tilt}} \sim (\lambda/4)L_{\text{mem}}$ with L_{mem} the length of the membranes. Detailed characterizations of such multi-membrane devices will be presented in future work. ...“

- in line 614, added some clarification for Equation 7:
“... and the Heaviside step function $\Theta(z)$...”
- in line 564, changed the format of an equation:
“... $\lambda=2\pi/k_0$...”
- in line 613, adjusted the formulation:
“... the explicit parametrization of the cavity ~~dielectric function~~ permittivity...”

REVIEWERS' COMMENTS

Reviewer #1 (Remarks to the Author):

The authors have done a good job at addressing the referees' comments within their current knowledge of the system, and, once again, this is high-quality work that certainly deserves publication. However, my concern regarding the lack of truly convincing evidence that high mechanical Qs can be achieved or the lack of a decisive optomechanics application pertains. I understand that the authors and some of the other referees may disagree, but, in my opinion, this makes this work fall short of publication in Nat. Comm.

Reviewer #2 (Remarks to the Author):

I think the authors have made a satisfactory reply to all of the referees' comments. I think the paper is suitable for publication.

Reviewer #3 (Remarks to the Author):

The authors have addressed remaining criticisms and improved the manuscript accordingly. In particular, the critical point on the limited mechanical quality factor and the perspective to use dissipation dilution is now described in detail. Also, the strength of 3D laser writing is now more explicitly demonstrated by adding a measurement of a two-membrane structure.

With these improvements, I now support publication of the manuscript.

We want to express again our thanks to all the reviewers for their careful review of our revised manuscript. We are grateful that all acknowledge that we have well-addressed their comments of the previous round. We are convinced that their feedback strongly contributed to a further improvement of the manuscript.

In the following, we want to reply to the assessment of each reviewer comments with the respective reviewers' comment formatted in *italic*.

With best regards,
the authors

Reviewer #1 (Remarks to the Author):

The authors have done a good job at addressing the referees' comments within their current knowledge of the system, and, once again, this is high-quality work that certainly deserves publication. However, my concern regarding the lack of truly convincing evidence that high mechanical Qs can be achieved or the lack of a decisive optomechanics application pertains. I understand that the authors and some of the other referees may disagree, but, in my opinion this makes this work fall short of publication in Nat. Comm.

We had hoped that the demonstration of a multi-mode mechanical structure, as it is now included in the manuscript after the previous reviewer feedback, would be a convincing application and therefore disagree with this part of the assessment. However, we are very glad that the referee acknowledges the improvements of the manuscript and states that it contains high-quality work.

Reviewer #2 (Remarks to the Author):

I think the authors have made a satisfactory reply to all of the referees' comments.

I think the paper is suitable for publication.

We are very happy that we were able to address the referee's comments to complete satisfaction.

Reviewer #3 (Remarks to the Author):

The authors have addressed remaining criticisms and improved the manuscript accordingly. In particular, the critical point on the limited mechanical quality factor and the perspective to use dissipation dilution is now described in detail. Also, the strength of 3D laser writing is now more explicitly demonstrated by adding a measurement of a two-membrane structure.

With these improvements, I now support publication of the manuscript.

We are pleased that the reviewer is satisfied by our incorporated changes and agree that the added material strongly improved the manuscript quality.